# Phosphonic Acid Analogs of Fluorophenylalanines as Inhibitors of Human and Porcine Aminopeptidases N: Validation of the Importance of the Substitution of the Aromatic Ring

**DOI:** 10.3390/biom10040579

**Published:** 2020-04-09

**Authors:** Weronika Wanat, Michał Talma, Błażej Dziuk, Jean-Luc Pirat, Paweł Kafarski

**Affiliations:** 1Department of Bioorganic Chemistry, Wroclaw University of Science and Technology, Wybrzeże Wyspiańskiego 27, 50-370 Wrocław, Poland; michal.talma@pwr.edu.pl (M.T.); blazej.dziuk@pwr.edu.pl (B.D.); pawel.kafarski@pwr.edu.pl (P.K.); 2Faculty of Chemistry, University of Opole, Oleska 48, 45-052 Opole, Poland; 3ICGM, University of Montpellier, ENSCM, CNRS, 34296 Montpellier, France; pirat@enscm.fr

**Keywords:** phosphonic acid analogs, human and porcine aminopeptidase, molecular modeling, fluorine, inhibitors

## Abstract

A library of phosphonic acid analogs of phenylalanine substituted with fluorine, chlorine and trifluoromethyl moieties on the aromatic ring was synthesized and evaluated for inhibitory activity against human (hAPN) and porcine (pAPN) aminopeptidases. Fluorogenic screening indicated that these analogs are micromolar or submicromolar inhibitors, both enzymes being more active against hAPN. In order to better understand the mode of the action of the most active compounds, molecular modeling was used. It confirmed that aminophosphonic portion of the enzyme is bound nearly identically in the case of all the studied compounds, whereas the difference in activity results from the placement of aromatic side chain of an inhibitor. Interestingly, both enantiomers of the individual compounds are usually bound quite similarly.

## 1. Introduction

Aminopeptidases are a heterogeneous group of enzymes that catalyze the controlled hydrolysis of N-terminal residues from proteins or peptide substrates. Thus, they play pivotal roles in regulation of protein turnover in nearly all organisms [1,2,3,4]. Many but not all of these enzymes are zinc dependent. Inhibition of a certain aminopeptidase activity usually results in profound effects on cell survival and proliferation. This suggests that aminopeptidases should be considered attractive targets for combating devastating human diseases; for instance, cancer, malaria or bacterial infections [5,6,7,8].

Zinc-dependent mammalian aminopeptidase N (alanine aminopeptidase, APN/CD13) is a type II trans-membrane enzyme belonging to M1 of the MA clan. APN mainly cleaves the neutral amino acid from N-terminus of peptides and proteins. Consequently, it plays multifunctional roles in important biological processes, including antigen processing and presentation; tumor invasion and metastasis; cell adhesion and motility; and neurotransmitter degradation [9,10,11,12]. It also serves as a receptor for entry of certain coronaviruses [13,14]. Since it was the first metallo-aminopeptidase to be biochemically and enzymatically characterized, it is perhaps the most extensively studied member in this family. Critical relevance of its activity for cancer progression caused intensive studies to be undertaken to develop new drugs directed towards this enzyme, including enzyme inhibitors and APN-targeted carrier constructs [15,16]. As a consequence, a variety of APN inhibitors have been developed [17,18,19,20,21,22,23]. The most successful example of this approach is bestatin, which is capable of inhibiting the growth of several tumors and is clinically used in the treatment with acute adult nonlymphocytic leukemia [24]. Since its discovery in 1974, it has remained the only inhibitor of APN on the market. Thus, to unlock the therapeutic potential of APN inhibitors, the design and synthesis of novel potent and selective inhibitors is required.

Since phenylalanine is quite commonly used scaffold in aminopeptidase inhibitors, we report the identification and characterization of a set of low-molecular phosphonic analogs of this amino acid as inhibitors of porcine and human enzymes. Phosphonic analog of phenylalanine (compound **1a**) is a micromolar inhibitor of porcine aminopeptidase N [25]. This work is a further development of the structural modulation of this molecule by studying the influences of substitutions of its aromatic ring with fluorine atoms on the inhibitory activity. Fluorine is considered to be a good mimic of hydrogen in medicinal chemistry, since the carbon-to-fluorine bond is only slightly (about 20%) longer than the carbon-to-hydrogen bond, and thus substitution of hydrogen by fluorine is well tolerated by a variety of proteins [26]. On the other hand, the electron-withdrawing nature of fluorine significantly affects electrostatic interactions. Additionally, it is well known that the introduction of fluorine atoms usually positively affects nearly all properties of the drug, and thus, today approximately 20–30% of pharmaceuticals on the marked are fluorinated molecules [27]. The favorable effect of the substitution of hydrogen for fluorine in the case of aminopeptidase inhibitors has been recently proven by our studies on phenylglycine analogs [28]. Thus, this report might be considered as an extension of that study. It describes far more effective inhibitors with non-trivial modes of binding, as suggested by molecular modelling.

## 2. Materials and Methods

### 2.1. Chemistry

All chemicals were purchased from commercial suppliers (Sigma Aldrich, Poznan, Poland; Trimen Chemicals, Lodz, Poland; POCh, Gliwice, Poland), were of analytical grade and were used without further purification. Column chromatography was performed using silica gel 60 (70–230 mesh). Analytical thin-layer chromatography was carried out on Merck SilicaGel 60 F254 (Darmstadt, Germany), and for visualization and detection of the compounds, UV light at 254 nm, ninhydrin and potassium permanganate were used. The ^1^H, ^13^C, ^19^F and ^31^P-NMR spectra were recorded on a Bruker Avance II Ultrashield Plus (Bruker, Rheinstetten, Germany) spectrometer operating at 400 MHz (^1^H), 101 MHz (^13^C), 376 MHz (^19^F) and 162 MHz (^31^P). Samples of the final products were diluted in the mixture of D_2_O+NaOD (99.8% at % D), with all solvents being supplied by ARMAR AG (Dottingen, Switzerland). Chemical shifts are reported relative to internal standards: TMS (^1^H NMR), CFCl_3_ (^19^F NMR) and 85% H_3_PO_4_ (^31^P NMR), and are given in parts per million (ppm), while coupling constant are reported in Hz. ^19^F NMR spectra were measured without decouplings with protons. Mass spectra were recorded at the Faculty of Chemistry, Wroclaw University of Science and Technology by using a Waters LCT Premier XE mass spectrometer (electrospray ionization, ESI) (Waters, Milford, MA, USA). Melting points were determined on an SRS Melting Point Apparatus OptiMelt MPA 100 (Stanford Research Systems, Sunnyvale, CA, USA) and reported uncorrected. All compounds were equimolar mixtures of R and S enantiomers. Analytical quality control of the final compounds was performed by HPLC (Chromservis, Praha, Czech Republic) (Kinetex 100A, C18, 5 µm, 150 mm × 4.6 mm) and was at least above 95%. A reverse phase column was used in combination with the following separation conditions: 99.95% H_2_O, 0.05% TFA (solvent A) and 99.5% CH_3_CN, 0.05% TFA (solvent B); 0.0–4.0 min, 0% B; 4.1–39.0 min 70% B. The flow rate was 1 mL/1 min and the UV signal was recorded at 220 nm.

### 2.2. Synthesis of the Fluorinated 1-Amino-2-phenylethylphosphonic Acids (1)–General Procedure

Fluorinated phenylacetic acid chlorides (**2**). The mixture of fluorinated phenylacetic acid (0.01 mol, 1 equiv) and thionyl chloride (0.04 mol, 4 equiv) was stirred at 80 °C for 2 h under atmosphere of nitrogen. After cooling to the room temperature, 20 mL of toluene was added and solvent was removed by distillation. This operation was repeated 4 times, until SOCl_2_ was completely removed.

Fluorinated diethyl keto/enol-2-phenylethylphosphonates (**3**/**4**). The obtained phenylacetyl chloride (**2**) was directly dissolved (without further purification) in dry diethyl ether (7 mL) and then cooled to 0 °C. During constant stirring, the solution of triethyl phosphite (1.99 g, 0.01 mol, 1 equiv) in 2 mL diethyl ether was added very slowly (the temperature of the reaction mixture cannot exceed 7–8 °C). After completion of the addition mixture was stirred 15–20 min in 0 °C and the solvent was removed under reduced pressure.

Fluorinated diethyl (E/Z)-1-hydroxyimino-2-phenylethylphosphonates (**5**). The obtained mixture of ketophosphonate and enolphosphonate (1 equiv) was dissolved, without further purification, in dry ethanol (2 mL) and added dropwise to the mixture of hydroxylamine hydrochloride (0.01 mol, 1 equiv) and 0.01 mol (1 equiv) of pyridine dissolved in 10 mL of dry ethanol. The reaction mixture was stirred at room temperature for 18–24 h. After this time, the volatile products were removed by rotary evaporation and the oily residue was treated with 5 mL of 0.5% HCl followed by extraction of the mixture 5 times with dichloromethane (5 × 10 mL). The organic phase was washed with 5% NaHCO_3_ and water, and was dried over anhydrous sodium sulfate. The crude product was purified by column chromatography (cyclohexane/ethyl acetate 1:1, *v*/*v*).

Fluorinated diethyl 1-amino-2-phenylethylphosphonates (**6**) and diethyl 1-N-formylamino-2-phenylethylphosphonates (**7**). The mixture of E/Z oxime (1 equiv) was dissolved in dry MeOH (10 mL) and ammonium formate (2 equiv) was added. Then, to the stirred mixture at reflux temperature, zinc dust (1 equiv) was added in small portions. The mixture was intensively stirred during 18–24 h at 68 °C, and the progress of the reaction was monitored by TLC (ninhydrin visualization) and ^31^P NMR. After the completion of the reaction, the mixture was filtered through celite and washed by methanol. The filtrate was evaporated under vacuum and the oily residues was taken into DCM, washed twice with 80% saturated brine to remove excess of ammonium formate and finally washed with water. The organic layer was dried over anhydrous Na_2_SO_4_ and evaporated. The obtained oil was purified by column chromatography (ethyl acetate containing 5% of ethanol) and the resulting aminophoshonate (**6**) was used in the next step.

Fluorinated 1-amino-2-phenylethylphosphonic acids (**1**). The aminophosphonate (**6**) was dissolved in 20 mL of 6 M HCl and refluxed 8 h. After the removal of the volatile products, the precipitated of α-aminophosphonic acid was filtered off and recrystallized from the mixture of water and ethanol.

#### 2.2.1. 1-Amino-2-phenylethylphosphonic Acid (**1a**) Was Obtained after Deiodination during Reduction of the Oxime Intermediate to Aminophosphonate Ester

White solid, m.p. 286 °C; yield: 12%; ^1^H NMR (400 MHz, D_2_O+NaOD) δ, ppm: 7.24–7.08 (m, 5H, 5xCH_ar_), 4.68 (br s, 1H, NH), 3.04 (ddd, *J* = 14.0, 5.0, 2.6 Hz, 1H), 2.32 (ddd, *J* = 14.0, 12.1, 6.0 Hz, 1H) (CH_2_), 2.69 (td, *J* = 11.7, 2.6 Hz, 1H, CHP); ^13^C NMR (101 MHz, D_2_O+NaOD) δ, ppm: 141.13 (d, *J* = 15.4 Hz, C_ar_), 129.26 (s, 2xC_ar_), 128.53 (s, 2xC_ar_), 126.15 (s, C_ar_), 52.15 (d, *J* = 138.7 Hz, CHP), 38.13 (s, CH_2_); ^31^P NMR (162 MHz, D_2_O+NaOD) δ, ppm: 20.95; HRMS (ESI-MS) *m*/*z* [MH]^+^ calculated for C_8_H_11_FNO_3_P: 288.0413, found: 288.0421.

#### 2.2.2. 1-Amino-2-(2-fluorophenyl)ethylphosphonic Acid (**1b**)

White solid, m.p. 253–255 °C; yield: 30%; ^1^H NMR (400 MHz, D_2_O+NaOD) δ, ppm: 7.26 (td, *J* = 7.7, 1.7 Hz, 1H, CH_ar_), 7.18 (tdd, *J* = 7.4, 5.4, 1.8 Hz, 1H, CH_ar_), 7.09–6.99 (m, 2H, 2xCH_ar_), 4.67 (br s, 1H, NH), 3.05 (d, *J* = 14.0 Hz, 1H), 2.59–2.51 (m, 1H) (CH_2_), 2.78–2.71 (m, 1H, CHP); ^13^C NMR (101 MHz, D_2_O+NaOD) δ, ppm: 161.41 (d, *J* = 242.2 Hz, C_ar_), 131.69 (d, *J* = 5.1 Hz, C_ar_), 128.04 (d, *J* = 8.2 Hz, C_ar_), 127.65 (t, *J* = 15.7 Hz, C_ar_), 124.24 (d, *J* = 3.4 Hz, C_ar_), 115.16 (d, *J* = 22.3 Hz, C_ar_), 51.35 (d, *J* = 137.6 Hz, CHP), 31.20 (s, CH_2_); ^19^F NMR (376 MHz, D_2_O+NaOD) δ, ppm: −118.68 (dd, *J* = 10.2, 6.3 Hz, 1F); ^31^P NMR (162 MHz, D_2_O+NaOD) δ, ppm: 20.87; HRMS (ESI-MS) *m*/*z* [MH]^−^ calculated for C_8_H_11_FNO_3_P: 218.0382, found: 218.0382.

#### 2.2.3. 1-Amino-2-(3-fluorophenyl)ethylphosphonic Acid (**1c**)

White solid, m.p. 258–260°C; yield: 32%; ^1^H NMR (400 MHz, D_2_O+NaOD) δ, ppm: 7.21 (dd, *J* = 14.5, 7.5 Hz, 1H, CH_ar_), 6.98 (dd, *J* = 19.5, 9.0 Hz, 2H, 2x CH_ar_), 6.86 (td, *J* = 9.2, 2.4 Hz, 1H, CH_ar_), 4.67 (br s, 1H, NH), 3.07 (ddd, *J* = 14.0, 4.3, 2.6 Hz, 1H), 2.45–2.29 (m, 1H) (CH_2_), 2.71 (td, *J* = 11.6, 2.5 Hz, 1H, CHP); ^13^C NMR (101 MHz, D_2_O+NaOD) δ, ppm: 162.67 (d, *J* = 242.4 Hz, C_ar_), 143.87 (dd, *J* = 15.6, 7.4 Hz, C_ar_), 129.95 (d, *J* = 8.5Hz, C_ar_), 125.06 (d, *J* = 2.5 Hz, C_ar_), 115.74 (d, *J* = 20.8 Hz, C_ar_), 112.73 (d, *J* = 21.1 Hz, C_ar_), 52.04 (d, *J* = 138.2 Hz, CHP), 37.91 (s, CH_2_); ^19^F NMR (376 MHz, D_2_O+NaOD) δ, ppm: −114.56 (td, *J* = 9.8, 6.4 Hz, 1F); ^31^P NMR (162 MHz, D_2_O+NaOD) δ, ppm: 20.70; HRMS (ESI-MS) *m*/*z* [MH]^−^ calculated for C_8_H_11_FNO_3_P: 218.0382, found: 218.0382.

#### 2.2.4. 1-Amino-2-(4-fluorophenyl)ethylphosphonic Acid (**1d**)

White solid, m.p. 287–290 °C; yield: 28%; ^1^H NMR (400 MHz, D_2_O+NaOD) δ, ppm: 7.19 (dd, *J* = 8.4, 5.8 Hz, 2H, CH_ar_), 6.96 (t, *J* = 8.8 Hz, 2H, CH_ar_), 4.72 (br s, 1H, NH), 3.02 (ddd, *J* = 14.1, 4.3, 2.8 Hz, 1H), 2.31 (ddd, *J* = 14.1, 12.1, 6.0 Hz, 1H) (CH_2_), 2.65 (ddd, *J* = 12.0, 11.1, 2.7 Hz, 1H, CHP); ^13^C NMR (101 MHz, D_2_O+NaOD) δ, ppm: 161.19 (d, *J* = 240.5 Hz, C_ar_), 136.77 (dd, *J* = 15.6, 3.0 Hz, C_ar_), 130.58 (d, *J* = 8.0 Hz, 2xC_ar_), 114.91 (d, *J* = 21.1 Hz, 2xC_ar_), 52.20 (d, *J* = 138.5, 0.8 Hz, CHP), 37.27 (s, CH_2_); ^19^F NMR (376 MHz, D_2_O+NaOD) δ, ppm: −118.19–−118.27 (m, 1F); ^31^P NMR (162 MHz, D_2_O+NaOD) δ, ppm: 20.87; HRMS (ESI-MS) *m*/*z* [MH]^−^ calculated for C_8_H_11_FNO_3_P: 218.0382, found: 218.0381.

#### 2.2.5. 1-Amino-2-(2,4-difluorophenyl)ethylphosphonic Acid (**1e**)

White solid, m.p. 286–288 °C; yield: 36%; ^1^H NMR (400 MHz, D_2_O+NaOD) δ, ppm: 7.18 (dd, *J* = 15.5, 8.5 Hz, 1H, CH_ar_), 6.83–6.70 (m, 2H, 2xCH_ar_), 4.68 (br s, 1H, NH), 2.95 (d, *J* = 14.1 Hz, 1H), 2.52–2.40 (m, 1H) (CH_2_), 2.64 (ddd, *J* = 13.0, 10.6, 2.7 Hz, 1H, CHP); ^13^C NMR (101 MHz, D_2_O+NaOD) δ, ppm: 161.15 (ddd, *J* = 243.8, 15.8, 11.8 Hz, 2xC_ar_), 132.14 (dd, *J* = 9.7, 6.7 Hz, C_ar_), 123.45 (td, *J* = 15.9, 3.7 Hz, C_ar_), 110.96 (dd, *J* = 20.9, 3.7 Hz, C_ar_), 103.31 (dd, *J* = 26.7, 25.5 Hz, C_ar_), 51.22 (d, *J* = 137.4 Hz, CHP), 30.66 (s, CH_2_); ^19^F NMR (376 MHz, D_2_O+NaOD) δ, ppm: −114.27–−114.36 (m, 1F), −114.61 (dd, *J* = 16.6, 8.7 Hz, 1F); ^31^P NMR (162 MHz, D_2_O+NaOD) δ, ppm: 20.78; HRMS (ESI-MS) *m*/*z* [MH]^+^ calculated for C_8_H_10_F_2_NO_3_P: 238.0445, found: 238.0456.

#### 2.2.6. 1-Amino-2-(2,5-difluorophenyl)ethylphosphonic Acid (**1f**)

White solid, m.p. 290–293 °C; yield: 34%; ^1^H NMR (400 MHz, D_2_O+NaOD) δ, ppm: 7.00–6.89 (m, 2H, 2xCH_ar_), 6.82 (ddd, *J* = 8.9, 7.7, 3.6 Hz, 1H, CH_ar_), 4.68 (br s, 1H, NH), 2.96 (dt, *J* = 14.0, 2.2 Hz, 1H), 2.56–2.41 (m, 1H) (CH_2_), 2.68 (ddd, *J =* 12.5, 10.2, 2.7 H, 1H, CHP); ^13^C NMR (101 MHz, D_2_O+NaOD) δ, ppm: 157.91 (ddd, *J* = 101.9, 100.2, 2.2 Hz, 2xC_ar_), 129.44 (ddd, *J* = 18.8, 15.6, 7.9 Hz, C_ar_), 117.48 (dd, *J* = 23.8, 5.4 Hz, C_ar_), 116.10 (dd, *J* = 25.7, 9.1 Hz, C_ar_), 113.99 (dd, *J* = 24.2, 8.8 Hz, C_ar_), 51.14 (d, *J* = 137.3 Hz, CHP), 31.26 (s, CH_2_); ^19^F NMR (376 MHz, D_2_O+NaOD) δ, ppm: −120.38 (dtd, *J* = 13.4, 8.6, 4.6 Hz, 1F), −124.73 (ddt, *J* = 18.5, 9.4, 4.8 Hz, 1F); ^31^P NMR (162 MHz, D_2_O+NaOD) δ, ppm: 20.59; HRMS (ESI-MS) *m*/*z* [MH]^−^ calculated for C_8_H_10_F_2_NO_3_P: 236.0284, found: 236.0288.

#### 2.2.7. 1-Amino-2-(2,6-difluorophenyl)ethylphosphonic Acid (**1g**)

White solid, m.p. 277–280 °C; yield: 32%; ^1^H NMR (400 MHz, D_2_O+NaOD) δ, ppm: 7.11 (tt, *J* = 8.4, 6.6 Hz, 1H, CH_ar_), 6.82 (t, *J* = 8.1 Hz, 2H, 2x CH_ar_), 4.67 (br s, 1H, NH), 3.02–2.88 (m, 1H,), 2.75–2.52 (m, 2H, CH_2_ and CHP); ^13^C NMR (101 MHz, D_2_O+NaOD) δ, ppm: 161.68 (dd, *J* = 244.0, 9.2 Hz, 2xC_ar_), 127.93 (t, *J* = 10.4 Hz, C_ar_), 116.03 (d, *J* = 15.4 Hz, C_ar_), 111.16–110.89 (m, 2xC_ar_), 50.68 (d, *J* = 136.7 Hz, CHP), 24.62 (d, *J* = 2.2 Hz, CH_2_); ^19^F NMR (376 MHz, D_2_O+NaOD) δ, ppm: −115.69 (t, *J* = 6.8 Hz, 2F); ^31^P NMR (162 MHz, D_2_O+NaOD) δ, ppm: 20.92 (d, *J*_PF_ = 0.6 Hz, 1P); HRMS (ESI-MS) *m*/*z* [MH]^−^ calculated for C_8_H_10_F_2_NO_3_P: 238.0445, found: 238.0455.

#### 2.2.8. 1-Amino-2-(3,4-difluorophenyl)ethylphosphonic Acid (**1h**)

White solid, m.p. 280–282 °C; yield: 28%; ^1^H NMR (400 MHz, D_2_O+NaOD) δ, ppm: 7.08–7.01 (m, 2H, 2xCH_ar_), 6.82 6.94–6.90 (m, 1H, CH_ar_), 4.67 (br s, 1H, NH), 3.02 (ddd, *J* = 14.1, 4.0, 3.1 Hz, 1H), 2.31 (ddd, *J* = 14.1, 12.1, 6.0 Hz, 1H) (CH_2_), 2.65 (ddd, *J* = 12.1, 11.2, 2.7 Hz, 1H, CHP); ^13^C NMR (101 MHz, D_2_O+NaOD) δ, ppm: 149.10 (ddd, *J* = 135.5, 123.8, 12.7 Hz, 2xC_ar_), 138.22 (ddd, *J* = 15.9, 5.7, 3.7 Hz, C_ar_), 125.26 (dd, *J* = 6.3, 3.3 Hz, C_ar_), 117.59 (d, *J* = 16.6 Hz, C_ar_), 116.83 (d, *J* = 16.9 Hz, C_ar_), 52.03 (d, *J* = 137.9 Hz, CHP), 37.36 (s, CH_2_); ^19^F NMR (376 MHz, D_2_O+NaOD) δ, ppm: −139.68–−139.80 (m, 1F), −143.25 (dddd, *J* = 15.1, 11.8, 7.9, 4.3 Hz, 1F); ^31^P NMR (162 MHz, D_2_O+NaOD) δ, ppm: 20.62; HRMS (ESI-MS) *m*/*z* [MH]^−^ calculated for C_8_H_10_F_2_NO_3_P: 236.0288, found: 236.0295.

#### 2.2.9. 1-Amino-2-(3,5-difluorophenyl)ethylphosphonic Acid (**1i**)

White solid, m.p. 265–267 °C; yield: 27%; ^1^H NMR (400 MHz, D_2_O+NaOD) δ, ppm: 6.80–6.74 (dd, *J* = 8.7, 2.0 Hz, 2H, 2xCH_ar_), 6.64 (tt, *J* = 9.5, 2.3 Hz, 1H, CH_ar_), 4.68 (br s, 1H, NH), 3.03 (ddd, *J* = 14.1, 4.9, 2.6 Hz, 1H), 2.34 (ddd, *J* = 14.1, 12.2, 6.0 Hz, 1H) (CH_2_), 2.68 (ddd, *J* = 12.0, 11.1, 2.7 Hz, 1H, CHP); ^13^C NMR (101 MHz, D_2_O+NaOD) δ, ppm: 162.72 (dd, *J* = 244.4, 11.5 Hz, C_ar_), 145.28 (s, C_ar_), 111.87 (d, *J* = 22.7 Hz, 2xC_ar_), 101.24 (t, *J* = 25.4 Hz, C_ar_), 51.90 (d, *J* = 137.2 Hz, CHP), 38.01 (s, CH_2_); ^19^F NMR (376 MHz, D_2_O+NaOD) δ, ppm: −111.66 (t, *J* = 8.9 Hz, 2F); ^31^P NMR (162 MHz, D_2_O+NaOD) δ, ppm: 20.45; HRMS (ESI-MS) *m*/*z* [MH]^−^ calculated for C_8_H_10_F_2_NO_3_P: 236.0288, found: 236.0295.

#### 2.2.10. 1-Amino-2-(2,4,5-trifluorophenyl)ethylphosphonic Acid (**1j**)

White solid, m.p. 275–277 °C; yield: 24%; ^1^H NMR (400 MHz, D_2_O+NaOD) δ, ppm: 7.12 (ddd, *J* = 11.1, 9.0, 7.0 Hz, 1H, CH_ar_), 6.97–6.90 (m, 1H, CH_ar_), 4.67 (br s, 1H, NH), 2.94 (d, *J* = 14.1 Hz, 1H), 2.51–2.42 (m, 1H) (CH_2_) 2.65 (ddd, *J* = 13.1, 10.7, 2.7 Hz, 1H, CHP); ^13^C NMR (101 MHz, D_2_O+NaOD) δ, ppm: 156.21 (ddd, *J* = 241.1, 9.7, 2.1 Hz, C_ar_), 149.47–145.07 (m, 2xC_ar_), 124.44–123.98 (m, C_ar_), 118.67 (dd, *J* = 18.9, 6.3 Hz, C_ar_), 105.04 (dd, *J* = 29.2, 20.9 Hz, C_ar_), 51.13 (d, *J* = 137.1 Hz, CHP), 30.67 (s, CH_2_); ^19^F NMR (376 MHz, D_2_O+NaOD) δ, ppm: −120.13 (dt, *J* = 14.8, 7.5 Hz, 1F), −138.23 (ddd, *J* = 20.4, 11.4, 3.0 Hz, 1F), −144.58 (dddd, *J* = 22.1, 15.3, 11.2, 6.9 Hz, 1F); ^31^P NMR (162 MHz, D_2_O+NaOD) δ, ppm: 20.47; HRMS (ESI-MS) *m*/*z* [MH]^−^ calculated for C_8_H_10_F_2_NO_3_P: 254.0194, found: 254.0193.

#### 2.2.11. 1-Amino-2-(2,4,6-trifluorophenyl)ethylphosphonic Acid (**1k**)

White solid, m.p. 285–288 °C; yield: 25%; ^1^H NMR (400 MHz, D_2_O+NaOD) δ, ppm: 6.66 (dd, *J* = 9.2, 8.0 Hz, 2H, 2xCH_ar_), 4.68 (br s, 1H, NH), 2.94–2.89 (m, 1H), 2.65–2.51 (m, 2H, CH_2_ and CHP); ^13^C NMR (101 MHz, D_2_O+NaOD) δ, ppm: 162.92–159.48 (m, 3xC_ar_), 112.60–111.96 (m, C_ar_), 100.10–99.51 (m, 2xC_ar_), 50.59 (d, *J* = 136.8 Hz, CHP), 24.34 (s, CH_2_); ^19^F NMR (376 MHz, D_2_O+NaOD) δ, ppm: −112.24–−112.32 (m, 1F), −112.83 (t, *J* = 6.4 Hz, 2F); ^31^P NMR (162 MHz, D_2_O+NaOD) δ, ppm: 20.60; HRMS (ESI-MS) *m*/*z* [MH]^−^ calculated for C_8_H_10_F_2_NO_3_P: 254.0194, found: 254.0193.

#### 2.2.12. 1-Amino-2-(3,4,5-trifluorophenyl)ethylphosphonic Acid (**1l**)

White solid, m.p. 270–272 °C; yield: 21%; ^1^H NMR (400 MHz, D_2_O+NaOD) δ, ppm: 6.90 (dd, *J* = 9.0, 6.8 Hz, 2H, 2xCH_ar_), 4.68 (br s, 1H, NH), 3.01 (dt, *J* = 14.3, 3.1 Hz, 1H), 2.31 (ddd, *J* = 14.2, 12.1, 6.0 Hz, 1H) (CH_2_), 2.65 (td, *J* = 12.0, 2.7 Hz, 1H, CHP); ^13^C NMR (101 MHz, D_2_O+NaOD) δ, ppm: 150.55 (ddd, *J* = 245.9, 9.8, 4.1 Hz, 2xC_ar_), 137.78 (dt, *J* = 245.4, 15.6 Hz, C_ar_), 137.84–137.48 (m, C_ar_), 112.97 (dd, *J* = 15.3, 5.2 Hz, 2xC_ar_), 51.86 (d, *J* = 137.7 Hz, CHP), 37.60 (s, CH_2_); ^19^F NMR (376 MHz, D_2_O+NaOD) δ, ppm: −136.61–−136.73 (m, 2F), −165.57–−165.72 (m, 1F); ^31^P NMR (162 MHz, D_2_O+NaOD) δ, ppm: 20.34; HRMS (ESI-MS) *m*/*z* [MH]^+^ calculated for C_8_H_10_F_2_NO_3_P: 256.0350, found: 256.0343.

#### 2.2.13. 1-Amino-2-(2,3,4,5,6-pentafluorophenyl)ethylphosphonic Acid (**1m**)

White solid, m.p. 291–293 °C; yield: 15%; ^1^H NMR (400 MHz, D_2_O+NaOD) δ, ppm: 4.68 (br s, 1H, NH), 3.00 (t, *J* = 12.5 Hz, 1H), 2.68–2.58 (m, 2H, CH_2_ and CHP); ^13^C NMR (101 MHz, D_2_O+NaOD) δ, ppm: 145.30 (dddd, *J* = 242.6, 12.5, 9.0, 3.4 Hz, 2xC_ar_), 140.77–140.36 (m, C_ar_), 138.71–137.89 (m, C_ar_), 136.29–135.70 (m, C_ar_), 113.89 (qd, *J* = 19.2, 3.3 Hz, C_ar_), 50.49 (d, *J* = 136.1 Hz, CHP), 24.89 (s, CH_2_); ^19^F NMR (376 MHz, D_2_O+NaOD) δ, ppm: −143.83 (dd, *J* = 23.0, 7.7 Hz, 2F), −158.83 (t, *J* = 21.0 Hz, 1F), −164.00 (dt, *J* = 22.7, 7.7 Hz, 2F); ^31^P NMR (162 MHz, D_2_O+NaOD) δ, ppm: 19.96; HRMS (ESI-MS) *m*/*z* [MH]^+^ calculated for C_8_H_10_F_2_NO_3_P: 292.0162, found: 292.0170.

#### 2.2.14. 1-Amino-2-(3-chloro-2-fluorophenyl)ethylphosphonic Acid (**1n**)

White solid, m.p. 284–285 °C; yield: 21%; ^1^H NMR (400 MHz, D_2_O+NaOD) δ, ppm: 7.13 (dt, *J* = 27.0, 6.8 Hz, 2H, 2xCH_ar_), 6.94 (t, *J* = 7.8 Hz, 1H, CH_ar_), 4.69 (br s, 1H, NH), 2.98 (d, *J* = 13.9 Hz, 1H), 2.49 (td, *J* = 13.1, 6.2 Hz, 1H) (CH_2_), 2.68–2.62 (m, 1H, CHP); ^13^C NMR (101 MHz, D_2_O+NaOD) δ, ppm: 156.47 (d, *J* = 244.1 Hz, C_ar_), 130.07 (d, *J* = 4.6 Hz, C_ar_), 129.40 (t, *J* = 15.7 Hz, C_ar_), 128.34 (s, C_ar_), 124.68 (d, *J* = 4.5 Hz, C_ar_), 120.01 (d, *J* = 18.4 Hz, C_ar_), 51.20 (d, *J* = 137.4 Hz, CHP), 31.43 (d, *J* = 1.4 Hz, CH_2_); ^19^F NMR (376 MHz, D_2_O+NaOD) δ, ppm: −121.14 (t, *J* = 6.8 Hz, 1F); ^31^P NMR (162 MHz, D_2_O+NaOD) δ, ppm: 20.59; HRMS (ESI-MS) *m*/*z* [MH]^−^ calculated for C_8_H_10_F_2_NO_3_P: 251.9993, found: 252.0000.

#### 2.2.15. 1-Amino-2-(4-chloro-2-fluorophenyl)ethylphosphonic Acid (**1o**)

White solid, m.p. 269–271 °C; yield: 29%; ^1^H NMR (400 MHz, D_2_O+NaOD) δ, ppm: 7.18 (t, *J* = 8.4 Hz, 1H, CH_ar_), 7.06–7.02 (m, 2H, 2xCH_ar_), 4.67 (br s, 1H, NH), 2.97 (d, *J* = 14.1 Hz, 1H), 2.52–2.44 (m, 1H) (CH_2_), 2.67 (ddd, *J* = 14.4, 10.4, 2.7 Hz, 1H, CHP); ^13^C NMR (101 MHz, D_2_O+NaOD) δ, ppm: 161.09 (d, *J* = 245.9 Hz, C_ar_), 132.43 (d, *J* = 6.0 Hz, C_ar_), 131.75 (d, *J* = 10.6 Hz, C_ar_), 126.92–126.06 (m, C_ar_), 124.30 (d, *J* = 3.5 Hz, C_ar_), 115.68 (d, *J* = 26.3 Hz, C_ar_), 51.14 (dd, *J* = 137.3, 0.7 Hz, CHP), 30.83 (d, *J* = 1.4 Hz, CH_2_); ^19^F NMR (376 MHz, D_2_O+NaOD) δ, ppm: −115.95 (t, *J* = 9.1 Hz, 1F); ^31^P NMR (162 MHz, D_2_O+NaOD) δ, ppm: 20.67; HRMS (ESI-MS) *m*/*z* [MH]^−^ calculated for C_8_H_10_F_2_NO_3_P: 251.9993, found: 252.0000.

#### 2.2.16. 1-Amino-2-(5-chloro-2-fluorophenyl)ethylphosphonic Acid (**1p**)

White solid, m.p. 279–281 °C; yield: 28%; ^1^H NMR (400 MHz, D_2_O+NaOD) δ, ppm: 7.24 (dd, *J* = 6.5, 2.7 Hz, 1H, CH_ar_), 7.10 (ddd, *J* = 8.7, 4.4, 2.7 Hz, 1H, CH_ar_), 6.93 (t, *J* = 8.9 Hz, 1H, CH_ar_), 4.68 (br s, 1H, NH), 2.97 (dt, *J* = 14.0, 2.2 Hz, 1H), 2.51–2.43 (m, 1H) (CH_2_), 2.67 (ddd, *J* = 13.1, 10.8, 2.6 Hz, 1H, CHP); ^13^C NMR (101 MHz, D_2_O+NaOD) δ, ppm: 159.99 (d, *J* = 242.3 Hz, C_ar_), 131.04 (d, *J* = 5.4 Hz, C_ar_), 129.62 (dd, *J* = 18.0, 15.7 Hz, C_ar_), 128.30 (d, *J* = 3.1 Hz, C_ar_), 127.60 (d, *J* = 8.7 Hz, C_ar_), 116.55 (d, *J* = 24.6 Hz, C_ar_), 51.13 (d, *J* = 137.5 Hz, CHP), 31.14 (d, *J* = 1.8 Hz, CH_2_); ^19^F NMR (376 MHz, D_2_O+NaOD) δ, ppm: −121.23–−121.30 (m, 1F); ^31^P NMR (162 MHz, D_2_O+NaOD) δ, ppm: 20.55; HRMS (ESI-MS) *m*/*z* [MH]^−^ calculated for C_8_H_10_F_2_NO_3_P: 251.9993, found: 251.9986.

#### 2.2.17. 1-Amino-2-(2-chloro-6-fluorophenyl)ethylphosphonic Acid (**1r**)

White solid, m.p. 278–281 °C; yield: 35%; ^1^H NMR (400 MHz, D_2_O+NaOD) δ, ppm: 7.12–7.04 (m, 2H, 2xCH_ar_), 6.92 (ddd, *J* = 9.5, 7.9, 1.6 Hz, 1H, CH_ar_), 4.68 (br s, 1H, NH), 3.06–2.97 (m, 1H), 2.84–2.71 (m, 2H, CH_2_ and CHP); ^13^C NMR (101 MHz, D_2_O+NaOD) δ, ppm: 161.72 (d, *J* = 244.5 Hz, C_ar_), 135.04 (d, *J* = 6.4 Hz, C_ar_), 128.17 (d, *J* = 9.8 Hz, C_ar_), 126.31 (dd, *J* = 19.1, 15.3 Hz, C_ar_), 125.19 (d, *J* = 3.3 Hz, C_ar_), 113.86 (d, *J* = 23.3 Hz, C_ar_), 50.28 (d, *J* = 136.4 Hz, CHP), 28.43 (d, *J* = 1.8 Hz, CH_2_); ^19^F NMR (376 MHz, D_2_O+NaOD) δ, ppm: −113.49(t, J = 7.5 Hz, 1F); ^31^P NMR (162 MHz, D_2_O+NaOD) δ, ppm: 20.73; HRMS (ESI-MS) *m*/*z* [MH]^−^ calculated for C_8_H_10_F_2_NO_3_P: 251.9993, found: 251.9995.

#### 2.2.18. 1-Amino-2-(4-chloro-3-fluorophenyl)ethylphosphonic Acid (**1s**)

White solid, m.p. 262–264 °C; yield: 31%; ^1^H NMR (400 MHz, D_2_O+NaOD) δ, ppm: 7.26 (t, *J* = 8.1 Hz, 1H, CH_ar_), 7.05 (dd, *J* = 10.6, 1.8 Hz, 1H, CH_ar_), 6.96 (dd, *J* = 8.2, 1.7 Hz, 1H, CH_ar_), 4.68 (br s, 1H, NH), 3.04 (ddd, *J* = 14.1, 4.9, 2.6 Hz, 1H), 2.34 (ddd, *J* = 14.1, 12.2, 6.0 Hz, 1H) (CH_2_), 2.67 (td, *J* = 12.0, 2.7 Hz, 1H, CHP); ^13^C NMR (101 MHz, D_2_O+NaOD) δ, ppm: 157.54 (d, *J* = 245.1 Hz, C_ar_), 142.45 (dd, *J* = 15.8, 6.8 Hz, C_ar_), 130.14 (s, C_ar_), 125.95 (d, *J* = 3.3 Hz, C_ar_), 117.38 (d, *J* = 17.6 Hz, C_ar_), 117.07 (d, *J* = 20.5 Hz, C_ar_), 51.93 (d, *J* = 137.9 Hz, CHP), 37.56 (d, *J* = 0.7 Hz, CH_2_); ^19^F NMR (376 MHz, D_2_O+NaOD) δ, ppm: −117.50 (dd, *J* = 10.5, 8.0 Hz, 1F); ^31^P NMR (162 MHz, D_2_O+NaOD) δ, ppm: 20.54; HRMS (ESI-MS) *m*/*z* [MH]^−^ calculated for C_8_H_10_F_2_NO_3_P: 251.9993, found: 251.9979.

#### 2.2.19. 1-Amino-2-(3-chloro-4-fluorophenyl)ethylphosphonic Acid (**1t**)

White solid, m.p. 284–287 °C; yield: 33%; ^1^H NMR (400 MHz, D_2_O+NaOD) δ, ppm: 7.27 (dd, *J* = 7.3, 2.1 Hz, 1H, CH_ar_), 7.08 (ddd, *J* = 7.1, 4.9, 2.1 Hz, 1H, CH_ar_), 7.04–7.00 (m, 1H, CH_ar_), 4.68 (br s, 1H, NH), 3.01 (ddd, *J* = 14.3, 4.0, 3.1 Hz, 1H), 2.30 (ddd, *J* = 14.2, 12.1, 6.0 Hz, 1H) (CH_2_), 2.65 (ddd, *J* = 12.0, 11.3, 2.7 Hz, 1H, CHP); ^13^C NMR (101 MHz, D_2_O+NaOD) δ, ppm: 156.27 (d, *J* = 243.2 Hz, C_ar_), 138.27 (dd, *J* = 15.8, 3.7 Hz, C_ar_), 130.81 (s, C_ar_), 129.11 (d, *J* = 7.2 Hz, C_ar_), 119.58 (d, *J* = 17.5 Hz, C_ar_), 116.21 (d, *J* = 20.8 Hz, C_ar_), 52.05 (dd, *J* = 137.9, 0.9 Hz, CHP), 37.16 (d, *J* = 0.7 Hz, CH_2_); ^19^F NMR (376 MHz, D_2_O+NaOD) δ, ppm: −121.08 (dd, *J* = 13.2, 8.3 Hz, 1F); ^31^P NMR (162 MHz, D_2_O+NaOD) δ, ppm: 20.59; HRMS (ESI-MS) *m*/*z* [MH]^−^ calculated for C_8_H_10_F_2_NO_3_P: 251.9993, found: 251.9975.

#### 2.2.20. 1-Amino-2-(2-chloro-4-fluorophenyl)ethylphosphonic Acid (**1u**)

White solid, m.p. 282–284 °C; yield: 32%; ^1^H NMR (400 MHz, D_2_O+NaOD) δ, ppm: 7.24 (dd, *J* = 8.6, 6.3 Hz, 1H, CH_ar_), 7.10 (dd, *J* = 9.0, 2.7 Hz, 1H, CH_ar_), 6.92 (td, *J* = 8.5, 2.7 Hz, 1H, CH_ar_), 4.67 (br s, 1H, NH), 3.09 (dt, *J* = 14.0, 3.6 Hz, 1H), 2.61 (ddd, *J* = 14.0, 12.3, 6.4 Hz, 1H) (CH_2_), 2.77 (ddd, *J* = 12.3, 10.5, 2.7 Hz, 1H, CHP); ^13^C NMR (101 MHz, D_2_O+NaOD) δ, ppm: 160.86 (d, *J* = 244.5 Hz, C_ar_), 134.41–134.20 (m, 2xC_ar_), 132.44 (d, *J* = 8.8 Hz, C_ar_), 115.09 (dd, *J* = 242.0, 22.8 Hz, 2xC_ar_), 50.93 (dd, *J* = 137.1, 0.9 Hz, CHP), 34.73 (d, *J* = 1.7 Hz, CH_2_); ^19^F NMR (376 MHz, D_2_O+NaOD) δ, ppm: −115.80 (td, *J* = 8.8, 6.5 Hz, 1F); ^31^P NMR (162 MHz, D_2_O+NaOD) δ, ppm: 20.89; HRMS (ESI-MS) *m*/*z* [MH]^−^ calculated for C_8_H_10_F_2_NO_3_P: 251.9993, found: 252.0002.

#### 2.2.21. 1-Amino-2-(2-trifluoromethylphenyl)ethylphosphonic Acid (**1v**)

White solid, m.p. 236–240 °C; yield: 24%; ^1^H NMR (400 MHz, D_2_O+NaOD) δ, ppm: 7.57 (d, *J* = 7.9 Hz, 1H, CH_ar_), 7.48–7.40 (m, 2H, 2xCH_ar_), 7.26 (t, *J* = 7.3 Hz, 1H, CH_ar_), 4.67 (br s, 1H, NH), 3.17 (d, *J* = 14.4 Hz, 1H), 2.72–2.63 (m, 1H) (CH_2_), 2.84–2.77 (m, 1H, CHP); ^13^C NMR (101 MHz, D_2_O+NaOD) δ, ppm: 139.50 (dq, *J* = 15.9, 1.5 Hz, C_ar_), 132.08 (d, *J* = 0.9 Hz, C_ar_), 131.50 (s, C_ar_), 128.47 (q, *J* = 29.2 Hz, C_ar_), 126.31 (s, C_ar_), 125.88 (q, *J* = 5.9 Hz, C_ar_), 124.74 (q, *J* = 273.5 Hz, CF_3_), 51.93 (d, *J* = 137.2 Hz, CHP), 33.96 (s, CH_2_); ^19^F NMR (376 MHz, D_2_O+NaOD) δ, ppm: −57.98 (s, 3F, CF_3_); ^31^P NMR (162 MHz, D_2_O+NaOD) δ, ppm: 20.87; HRMS (ESI-MS) *m*/*z* [MH]^−^ calculated for C_8_H_11_FNO_3_P: 270.0500, found: 270.0507.

#### 2.2.22. 1-Amino-2-(3-trifluoromethylphenyl)ethylphosphonic Acid (**1w**)

White solid, m.p. 250–252 °C; yield: 29%; ^1^H NMR (400 MHz, D_2_O+NaOD) δ, ppm: 7.52 (s, 1H, CH_ar_), 7.46–7.32 (m, 3H, 3xCH_ar_), 4.68 (br s, 1H, NH), 3.13 (ddd, J = 14.1, 4.8, 2.6 Hz, 1H), 2.43 (ddd, J = 14.0, 12.2, 6.0 Hz, 1H) (CH_2_), 2.73 (td, J = 11.7, 2.6 Hz, 1H, CHP); ^13^C NMR (101 MHz, D_2_O+NaOD) δ, ppm: 142.09 (d, J = 14.2 Hz, C_ar_), 132.98 (s, C_ar_), 129.79 (d, J = 32.6 Hz, C_ar_), 128.96 (s, C_ar_), 125.79 (s, C_ar_), 124.34 (q, J = 240.7 Hz, CF_3_), 122.88 (s, C_ar_), 52.06 (d, J = 138.3 Hz, CHP), 37.97 (s, CH_2_); ^19^F NMR (376 MHz, D_2_O+NaOD) δ, ppm: −62.03 (s, 3F, CF_3_); ^31^P NMR (162 MHz, D_2_O+NaOD) δ, ppm: 20.65; HRMS (ESI-MS) *m*/*z* [MH]^−^ calculated for C_8_H_11_FNO_3_P: 270.0500, found: 270.0507.

#### 2.2.23. 1-Amino-2-(4-trifluoromethylphenyl)ethylphosphonic Acid (**1x**)

White solid, m.p. 268–270 °C; yield: 23%; ^1^H NMR (400 MHz, D_2_O+NaOD) δ, ppm: 7.43 (dd, *J* = 70.2, 8.0 Hz, 1H, 4xCH_ar_), 4.68 (br s, 1H, NH), 3.12 (d, *J* = 13.3 Hz, 1H), 2.48–2.38 (m, 1H) (CH_2_), 2.73 (td, *J* = 11.8, 2.4 Hz, 1H, CHP); ^13^C NMR (101 MHz, D_2_O+NaOD) δ, ppm: 145.64 (d, *J* = 14.7 Hz, C_ar_), 129.93 (s, 2xC_ar_), 127.55 (q, *J* = 30.2 Hz, C_ar_), 125.96 (s, C_ar_), 125.21 (s, C_ar_), 124.54 (q, *J* = 271.5 Hz, CF_3_), 52.02 (d, *J* = 138.0 Hz, CHP), 38.06 (s, CH_2_); ^19^F NMR (376 MHz, D_2_O+NaOD) δ, ppm: −61.78 (s, 3F, CF_3_); ^31^P NMR (162 MHz, D_2_O+NaOD) δ, ppm: 20.68; HRMS (ESI-MS) *m*/*z* [MH]^−^ calculated for C_8_H_11_FNO_3_P: 270.0500, found: 270.0507.

#### 2.2.24. 1-Amino-2-(5-fluoro-2-trifluoromethylphenyl)ethylphosphonic Acid (**1y**)

White solid, m.p. 270–272 °C; yield: 23%; ^1^H NMR (400 MHz, D_2_O+NaOD) δ, ppm: 7.58 (dd, *J* = 8.5, 5.9 Hz, 1H, CH_ar_), 7.17 (d, *J* = 10.0 Hz, 1H, CH_ar_), 6.96 (t, *J* = 8.1 Hz, 1H, CH_ar_), 4.68 (br s, 1H, NH), 3.13 (d, *J* = 14.2 Hz, 1H), 2.81–2.63 (m, 1H, CH_2_ and CHP); ^13^C NMR (101 MHz, D_2_O+NaOD) δ, ppm: 164.35 (d, *J* = 249.4 Hz, C_ar_), 143.12 (s, C_ar_), 128.49 (s, C_ar_), 124.88 (q, *J* = 29.4 Hz, C_ar_), 124.42 (q, *J* = 272.7 Hz, CF_3_), 118.03 (d, *J* = 21.8 Hz, C_ar_), 113.18 (d, *J* = 22.4 Hz, C_ar_), 51.80 (d, *J* = 135.2 Hz, CHP), 33.99 (s, CH_2_); ^19^F NMR (376 MHz, D_2_O+NaOD) δ, ppm: −57.42 (s, 3F, CF_3_), −109.30 (s, 1F); ^31^P NMR (162 MHz, D_2_O+NaOD) δ, ppm: 20.54; HRMS (ESI-MS) *m*/*z* [MH]^+^ calculated for C_8_H_11_FNO_3_P: 288.0413, found: 288.0423.

#### 2.2.25. 1-Amino-2-(4-fluoro-3-trifluoromethylphenyl)ethylphosphonic Acid (**1z**)

White solid, m.p. 284–286 °C; yield: 19%; ^1^H NMR (400 MHz, D_2_O+NaOD) δ, ppm: 7.45 (dd, *J* = 31.0, 6.8 Hz, 2H, 2xCH_ar_), 7.12–7.07 (m, 1H, CH_ar_), 4.68 (br s, 1H, NH), 3.07 (d, *J* = 14.4 Hz, 1H), 2.39 (td, *J* = 13.1, 6.0 Hz, 1H) (CH_2_), 2.67 (t, *J* = 11.4 Hz, 1H, CHP); ^13^C NMR (101 MHz, D_2_O+NaOD) δ, ppm: 157.97 (dq, *J* = 250.4, 2.2 Hz, C_ar_), 137.34 (dd, *J* = 15.9, 3.7 Hz, C_ar_), 134.97 (dd, *J* = 8.4, 1.0 Hz, C_ar_), 127.52 (qd, *J* = 4.5, 1.3 Hz, C_ar_), 122.94 (qd, *J* = 271.3, 1.0 Hz, CF_3_), 117.42–116.32 (m,C_ar_), 116.54 (d, *J* = 20.4 Hz, C_ar_), 52.05 (dd, *J* = 137.9, 0.7 Hz, CHP), 37.19 (s, CH_2_); ^19^F NMR (376 MHz, D_2_O+NaOD) δ, ppm: −60.87 (s, 3F, CF_3_), −120.53 (s, 1F); ^31^P NMR (162 MHz, D_2_O+NaOD) δ, ppm: 20.54; HRMS (ESI-MS) *m*/*z* [MH]^+^ calculated for C_8_H_11_FNO_3_P: 288.0413, found: 288.0421.

#### 2.2.26. (1-Amino-3,3,3-trifluoropropyl)phosphonic acid (**8**)

(1-Amino-3,3,3-trifluoropropyl)phosphonic acid (**8**) was synthesized by using the procedure of amidoalkylation of phosphorous trichloride (Oleksyszyn reaction) modified by Ziora and Kafarski, based on Soroka procedure [29,30]. The final product was precipitated from its solution of hydrochloride in ethanol by addition of the pyridine to pH around 6. Pure product was recrystallized from mixture of water and ethanol.

White solid, m.p. 288–290 °C; yield: 11%; ^1^H NMR (400 MHz, D_2_O+NaOD) δ, ppm: 4.66 (br s, 1H, NH), 3.49 (ddd, *J* = 15.5, 10.9, 2.3 Hz, 1H, CHP), 2.83–2.49 (m, 2H, CH_2_). ^13^C NMR (101 MHz, D_2_O+NaOD) δ, ppm: 125.86 (qd, *J* = 276.7, 17.8 Hz, CF_3_), 43.35 (dd, *J* = 141.5, 2.4 Hz, CHP), 32.59 (q, *J* = 30.2 Hz, CF_3_CH_2_); ^19^F NMR (376 MHz, D_2_O+NaOD) δ, ppm: −64.56 (t, *J* = 10.8 Hz, 3F, CF_3_). ^31^P NMR (162 MHz, D_2_O+NaOD) δ, ppm: 11.05; HRMS (ESI-MS) *m*/*z* [MH]^+^ calculated for C_3_H_7_F_3_NO_3_P: 194.0112, found: 194.0134.

### 2.3. Bioassays

#### 2.3.1. Enzymatic Studies

Recombinant human alanine aminopeptidase (hAPN, EC 3.4.11.2) and porcine kidney alanine aminopeptidase (pAPN, EC 3.4.11.2) were purchased as lyophilized powder from R&D System (Minneapolis, MN, Canada) and Sigma Aldrich (Poznan, Poland), respectively. Fluorogenic substrate—Ala-AMC (l-alanine-4-methylcoumaryl-7-amide) was obtained from PeptaNova (Sandhausen, Germany). Both enzymes were dissolved in corresponding 50 mM Tris-HCl buffer and directly used in enzymatic analysis. The pHs of the buffers were as follows: 7.00 for hAPN and 7.20 for pAPN. Michaelis constants (Km) for the both enzymes towards Ala-AMC were determined as described in the literature [31]. Km values were 73 µM for hAPN and 90 µM for pAPN. Due to the very high polarity and low solubility of all the inhibitors in the assay buffer, they were transformed into hydrochlorides. Inhibitory constants measurements were carried out in 96-well plates, the total volume of 100 µL at 37 °C using a spectrofluorometer (SpectraMax Gemini EM fluorometer—Molecular Devices, San Jose, USA) operating at two wavelengths: excitation 355 nm and emission 460 nm. hAPN and pAPN were preincubated for 30 min in the assay buffer and further added to the wells containing appropriate concentrations of the substrate and the inhibitor. The measurement conditions in the plates were as follows: (i) eight inhibitor concentrations in the range 100–0.8 µM, (ii) substrate concentration of 50 µM for hAPN and 80 µM for pAPN and (iii) enzyme concentrations of 0.66 nM. The release of the fluorogenic leaving group of AMC was monitored for at least 15 min. The rate of enzymatic reaction was studied following the reaction progress curve, based on which velocity was calculated. Each experiment was repeated three times and standard deviation values were calculated. The type of the inhibition and inhibitory constants were calculated by using the Lineweaver–Burk, Dixon plot and Cheng–Prusoff equation: K_i_ = IC_50_/[1 + (S/K_m_)] as described earlier [32]. Kinetics parameters were calculated using GrafPad Prism and Microsoft Excel computer programs.

#### 2.3.2. Molecular Modelling

Crystal structures of enzymes were obtained from the Research Collaboratory for Structural Bioinformatics Protein Data Bank (RCSB-PDB): for human (*Homo sapiens*) M1 aminopeptidase 4FYT and porcine (*Sus scrofa*) M1 aminopeptidase 4FKE [10,33]. The proteins were protonated at experimental pH. Before docking, the structures of the inhibitors and their stereochemistry were considered, protonated in experimental pH typical for each enzyme also, and optimized by LigPrep [34].

The binding modes of the inhibitors of both aminopeptidases were studied by molecular modeling. All of the compounds (both enantiomers) were docked with the use of the induced fit docking algorithm of the Maestro Schrodinger package [35]. This algorithm indicated which of the amino acids were well scored, and these were therefore considered in the next step (see Appendix A in the Appendix A). The VSGb (variable-dielectric generalized Born) model was used, which incorporates residue-dependent effects. The solvent was water. Ligands were docked with the sample ring conformations option with a 2.5 kcal/mol energy window and standard glide, prime refinement and glide redocking (SP) procedures for the best pose for each compound [36]. MM-GBSA (molecular mechanics-generalized Born surface area) was performed as rePrime refinement to calculate Gibbs free energies with protein flexibility, with the distances from ligands also set as 0.0 Å and 5.0 Å. The first pose with the lowest binding energy in 5.0 Å was selected as the best one.

#### 2.3.3. Crystallography

The crystals of the diethyl 1-*N*-formylamino-2-(4-fluorophenyl)ethylphosphonate (compound 7d, Appendix A) were obtained after dissolution of chromatographically isolated crude product in ethanol and slow evaporation at room temperature. The single crystals were mounted on a CCD Xcalibur diffractometer (graphite monochromatic, MoKα radiation, λ = 0.71073 Å) at 100.0(1) K. The reciprocal space was explored by ω scans with detector positions at 60 mm distance from the crystal. The diffraction data processing of studied compounds (Lorentz and polarization corrections were applied) was performed using the CrysAlis CCD [37]. The structure was solved in the triclinic crystal system, P 1 space group (Appendix A), by direct methods and refined by a full-matrix least-squares method using SHELXL14 program [38,39]. The H atoms were located from difference Fourier synthesis and from geometrical parameters and refined using a riding model. The structure drawings were prepared using SHELXTL and Mercury programs [40].

Crystallographic data for solved structures have been deposited with the Cambridge Crystallographic Data Centre as supplementary publication number CCDC: 1963310. These data can be obtained free of charge via http://www.ccdc.cam.ac.uk/conts/retrieving.html, or from the Cambridge Crystallographic Data Centre, 12 Union Road, Cambridge CB2 1EZ, UK; fax: 144 1223 336 033; email: deposit@ccdc.cam.ac.uk.

## 3. Results and Discussion

### 3.1. Chemistry

Phosphonic acid analogs of phenylalanine (compound 1a), substituted with different numbers of fluorine atoms or trifluoromethyl groups (compounds 1) in the phenyl ring, have been chosen as presumable inhibitors of hAPN and pAPN. Since aminopeptidases essentially require a free amino group of the substrate or inhibitor, we could not use the recently described procedures for their preparation because these procedures deliver N-alkyl derivatives of compounds 1 [41].

The desired inhibitors had been synthesized by modified and optimized multistep reaction, described for the first time by Kowalik, et al. [42,43,44]. Bearing in mind the known instability of intermediates: phenylacetic acid chlorides, ketophosphonates and their oximes, we decided to apply the “on-line” process of using crude intermediates without purification (Scheme 1).

Briefly, the reactions began with the conversions of phenylacetic acids into their chlorides **2** by action of thionyl chloride. Resulting chlorides were immediately reacted with triethyl phosphite yielding the corresponding ketophosphonates (compounds **3**), which exist in equilibrium with tautomeric enols (compounds **4**) [45]. Subsequent reaction of these mixtures with hydroxylamine resulted in mixtures of E/Z oximes **5**. Reduction of the oximes, performed by using ammonium formate and zinc in molar ratio 2:1 [46], provided two phosphonates–compounds **6** and their formylated derivatives **7** clearly seen in ^31^P NMR spectra. We succeeded in the isolation and crystal structure determination of one of them; namely, diethyl 1-*N*-formylamino-2-(4-fluorophenyl)ethylphosphonate (compound **7d**, Appendix A). The formation of these side products is not surprising, and it does not have an influence on the overall yield of aminophosphonic acids **1**, which were obtained, after hydrolysis of the mixtures of compounds 6 and 7 with concentrated hydrochloric acid, with a satisfactory overall yield of about 30% (see representative NMR spectra, Appendix A). Reduction of oximes is a key step in this procedure, and the alternative application of ammonium formate/10% Pd/C [47] and ammonium formate/magnesium systems failed [48].

It is worth mentioning that we did not observe the formation of products of the recently reported intramolecular S_N_Ar reaction of compounds 6 possessing fluorine in the position *ortho* to the aminoethylphosphonate fragment. This reaction according to the literature should provide corresponding indolinylphosphonates [41].

The course of the applied procedure was studied in more detail in the case of preparation of 1-amino-(3,4,5-trifluorophenyl)ethylphosphonic acid (compound **1l**). Thus, ^1^H and ^31^P NMR spectra of the mixture of ketophosphonate 3l and enol 4l confirm the presence of the two tautomers being in equilibrium in a ratio of 66:34 (see Appendix A). This mixture was converted to isomeric E/Z oximes 5l in 54:46 ratio, which upon storage in CDCl_3_ converted into only one isomer (see Appendix A). Basing on the possible stabilization of *Z*-isomer by intramolecular hydrogen bonding (Scheme 2) we speculate that it is the final product. A similar effect was recently described for 1-methoxyindole-3-carboxaldehyde oxime [49].

We were also going to obtain compounds **1** substituted additionally with chlorine, bromine and iodine. These reactions carried out with substrates substituted with chlorine gave readily the desired products (compounds **1n**–**1u**). In the case of substrates substituted with bromine or iodine upon reduction of corresponding oximes, debromination and deiodination were observed (for a representative example see Appendix A). The reductive dehalogenation upon action of zinc powder in basic and acidic media was already described, and also includes the reaction carried out by zinc dust in the presence of ammonium formate and base (mainly NaOH) [50,51].

### 3.2. Enzymatic Studies

Aminopeptidase N from porcine kidney (pAPN) and human alanine aminopeptidase (hAPN) exhibit similar pattern of activity towards series of the same synthetic substrates and inhibitors. Thus, we performed medium-throughput screening on a collection of preselected novel analogs of phenylalanine towards pAPN and hAPN. This set of compounds was supplemented with 1-amino-3,3,3-trifluoropropylphosphonic acid (**8**). The latter one was obtained by standard amidoalkylation of 3,3,3-trifluoropropanal with acetamide and phosphorus trichloride [29,30]. All the compounds had been used as racemic mixtures.

The results of the structure–activity relationship exploration are presented in Table 1. With the exception of two compounds (**1v** and **1y**), which were practically inactive, all the remaining ones appeared to be micromolar or submicromolar inhibitors of both aminopeptidases. Because these molecules are smaller than typical drug molecules, the observed binding affinities may be considered substantial. Their potencies towards human enzymes are about an order in magnitude higher than for porcine enzymes. However, the pattern of structure–activity is nearly identical.

Phosphonic acid analog of phenylalanine (1-amino-2-phenylethylphosphonic acid, PheP, compound 1a) was treated as prototypical compound and may be considered positive control. Its inhibitory activity towards pAPN is in a good agreement with that found in the literature (73.3 μM for racemic mixture versus 27.5 μM found for l-aminophosphonate-PheP) [25]. Ten out of the 24 phosphonic analogs of phenylalanine displayed significantly higher activity than PheP towards both enzymes. In the case of human enzyme (hAPN) three of them (compounds **1i**, **1l** and **1t**) demonstrated submicromolar inhibitory constants. The remaining seven displayed only 2–10 fold weaker effects. Similarly potent activity was observed in the case of porcine enzyme (pAPN), for which compounds **1i**, **1l** and **1t** acted as micromolar inhibitors. On the other hand, six inhibitors conferred weaker activity than PheP. Aromatic rings of three of them were substituted with trifluoromethyl group (compounds **1v**, **1x** and **1y**). In this respect, the rather high potencies of compounds **1w** and **1z** are somewhat surprising. The remaining four compounds were more or less equipotent with the positive control **1a**. Compound **8** appeared to be a weak inhibitor of both enzymes.

Generally, substitution of the aromatic ring of analogs of phenylalanine with fluorine atoms is well accepted by both enzymes. The only exception is *perfluoro*-analog, which displays weak activity. Inspection of inhibitory constants reveals that substitution of the phenyl ring in *ortho* position is unfavorable, whereas substitution in positions *meta* and *para* are beneficial. This is also well seen in the case of compounds where fluorine was replaced by bulky chlorine or trifluoromethyl substituents. Comparison of activities of compounds **1n** and **p** versus compound **1r** or compound **1z** versus **1y** may serve as examples here.

Summing up, by substitution of phenyl ring of phosphonic analog of phenylalanine with fluorine and chlorine atoms, we have development new building blocks, which might be useful for design of novel, effective and perhaps selective inhibitors of aminopeptidases.

### 3.3. Molecular Docking

Computational approaches have become a key step in the rational development of new potential therapeutics and for the determination of binding modes of synthesized compounds. Thus, molecular modelling was performed to gain further insights into the complexation of compounds **1** by hAPN and pAPN and to determine the key factors responsible for that binding.

The aim of docking of compounds **1** was to reveal whether they exhibited similar binding to each other. First, we compared the binding cavities of both enzymes, which were found to be substantially similar in terms of architecture of the active site and revealed the presence of two similar, albeit different, hydrophobic binding pockets S1 and S1′ (Figure 1). The pockets of porcine enzyme are more spacious than human ones. Since enzymatic studies had been performed for racemic mixtures, two enantiomers of each compound **1** were docked to each enzyme. The computed induced fit docking algorithm indicated which of the amino acids were well scored in terms of coordination of inhibitors. Appendix A list the amino acids, which interact with arbitrary positions of both of enantiomers for each of the compound **1**. The most significant amino acids involved in the interaction with a certain inhibitor (those with the lowest energy of the enzyme-inhibitor complex) were marked green.

Computational analysis confirmed in vitro binding properties by demonstrating good surface complementarity of the inhibitors with the binding-pockets of the two aminopeptidases. Figure 1 presents global views of the results of docking of all the 25 analogs (both isomers) of phenylalanine (compounds **1a**–**1z**) to hAPN (Figure 1A) and pAPN (Figure 1B). These studies revealed an almost identical coordination mode of the aminophosphonate portions of all molecules to both enzymes. The phosphonate group is considered a transition-state analog of the hydrolysis of peptide bond, and its binding seems to confirm this assumption. Thus, two phosphonate oxygens are involved in zinc complexation, whereas a third oxygen forms a hydrogen bond with tyrosine (Tyr472 of pAPN and Tyr477 of hAPN). In most cases the free α-amino group shows a typical pattern of contacts with glutamate and glutamine residues (Glu350 and Glu384 of pAPN; and Glu355, Glu411 and Gln213 of hAPN). The hydrophobic S1 pocket of both aminopeptidases is responsible for recognition of N-terminal amino acids of the hydrolyzed substrates. Filling the S1 pocket with aromatic fragments of inhibitors should thus play a canonical role in inhibitory process. The molecular docking indicated that it indeed seems to be true for the human enzyme; however, in rare cases, the aromatic side-chains of the studied inhibitors interact preferably with the S1′ pocket, a pocket which is naturally involved in accommodating the C-terminal side of a hydrolyzed peptide substrate. In the case of the porcine enzyme, reversal of this situation is observed, since most of the inhibitors locate their aromatic parts in the S1′ pocket, which is a surprising and unexpected result. A similar effect, albeit less pronounced, was already observed earlier for interactions of phosphonic acid analogs of phenylglycine and pAPN [28].

The most important differences are seen between types and positions of the substituents in the aromatic moieties of the inhibitors, and their interactions with surrounding amino acid backbones. The nature of these interactions can significantly affect the arrangement of phenyl ring inside hydrophobic S1 and S1′ pockets and determine the final potency of the respective inhibitors. Thus, we docked the most active compounds (**1i**, **1l** and **1t**) and one of the less active compounds (**1g**) to hAPN and pAPN in order to understand their dominant interactions and to compare modes of binding with the two enzymes. In these cases, all modeling predicted the locations of all three inhibitors’ side chains to be in the S1 pocket of hAPN and in the S1′ pocket of pAPN.

Docking of both enantiomers of the most active 1-amino-2-(3,5-difluorophenyl)ethylphosphonic acid (compound **1i**, Figure 2) shows that the difference in binding of enantiomers of this compound by the two enzymes, although visible, is not significant. This effect was more pronounced for the human enzyme (Figure 2A), wherein the amino group from S isomer is coordinated by the side chain of Met354 of the S1 pocket, instead a typical contribution of the Glu411 in hydrogen binding acting as it is in the case of the R isomer. This difference should moderately affect the binding affinities of both enantiomers. In the case of porcine enzyme (Figure 2B) the differences in binding of the two enantiomers are significant and should reflect in their different affinities to the enzyme. This difference results from the lack of contacts of the aromatic fragment of the S isomer with the neighboring amino acids side chains, whereas the aromatic portion of the R isomer interacts with the hydrophobic S1′ pocket built of Gly347, Ala348, Ala361 and Val380.

The binding mode of 1-amino-2-(3,4,5-trifluorophenyl)ethylphosphonic acid (compound **1l**, Figure 3) is quite similar to that of **1i**. Computations predict similar binding modes of this compound to both enzymes. The differences in the geometric binding of R and S isomers could be considered minute, albeit their affinities should be different. In the case of hAPN (Figure 3A), additional coordination of the phosphonic moiety by Ala253 residue seems to be responsible by optimal arrangements of both isomers in the active site. In the case of the R isomer, this alanine additionally interacts with the amine portion of **1l**. The aromatic portion of isomer S is coordinated by π–π stacking with Phe896, while *para*-fluorine forms a hydrogen bond with Ser895. These interactions are anchoring this isomer in the S1 pocket and should be reflected in its stronger binding. In the case of pAPN (Figure 3B), alanine (Ala346) is involved in the interaction with the phosphonic group only of isomer R. This isomer is also additionally stabilized by interaction of its amine with His383, and thus should be a more effective inhibitor.

Quite interestingly, the docking predicted two slightly different modes of binding of both enantiomers of 1-amino-2-(3-chloro-4-fluorophenyl)ethylphosphonic acid (compound **1t**) to hAPN (Appendix A). Depending on the absolute configuration of the α-carbon of this inhibitor, the position of enzyme Phe472 is changed. This reflects with positioning of both enantiomers of the inhibitor. Thus, in the case of the R isomer, calculations predicted an interaction of Phe472 with the aromatic ring of the inhibitor, whereas in the case of the S isomer—the chlorine substituent. For isomer S, additional interactions between chlorine with Ala214 and fluorine with Ser895 contributed to slight diversification of its emplacement in hydrophobic pocket S1. It is worth noting that fluorine-Ser 895 interactions are identical for S isomers of compounds **1l** and **1t**, thereby explaining why this substitution is profitable.

In the case of pAPN (Appendix A), the types of the coordination of **1t** by amino acids side chains building the active site of the enzyme and the displacements of both enantiomers in the S1′ pocket may be considered identical. Substitution of the aromatic ring by chlorine and fluorine atoms increased the electron-rich character and improved the fundamental contacts with alanine and valine side chains, which are responsible for the most efficient docking in the hydrophobic pocket. Similarly to the case of compound **1l**, the amine group of **1t** forms an additional contact with His383.

Examination of the calculated pattern of binding of weak inhibitor 1-amino-2-(2,6-difluorophenyl)ethylphosphonic acid (compound **1g**) (Appendix A) surprisingly revealed very similar pattern of interactions of enantiomers of this compound to that observed in the case of compound **1t**. Once more, the phosphonic part of the molecule interacts with zinc ion and tyrosines present in the active site, whereas aromatic part of **1g** is bound in the S1 pocket of hAPN (Appendix A) and the S1′ pocket of pAPN (Appendix A). Moreover, fluorine atoms are involved in hydrogen bonds with some portions of the enzyme (Gln213 and Glu389 in the case of hAPN and Ala346 for pAPN). Thus, the pattern of binding does not explain the huge (two orders in magnitude) differences in affinities of **1t** versus **1g** observed experimentally. However, calculations indicate that there exists strong internal hydrogen bonding between fluorine and amine nitrogen atoms in molecule **1g** (Appendix A), which has to be disrupted upon binding. This might affect the energetics of the binding similarly to that reported for the influence the desolvation of peptidomimetic inhibitors, which strongly decreases their affinities towards thermolysin [52].

Summing up, the major driving force of the binding of compounds **1** by both enzymes is constituted by the interactions of the aminophosphonic portions of the inhibitors, which takes place in the active sites of the enzymes. Although modeling predicts that aromatic side chains of the studied inhibitors are bound in two different hydrophobic pockets (S1 in case of hAPN and S1′ for pAPN) the structure–activity relationship is similar for both enzymes. This suggests that S1 and S1′ pockets are similar to each other. On the other hand, the bindings of the aromatic moieties of **1i**, **1l** and **1t** to the S1 pocket of hAPN versus the S1′ pocket of pAPN seem to explain the differences in affinities of these compounds towards both enzymes.

## 4. Conclusions

Phosphonic acid analogs of phenylalanine substituted in their respective phenyl rings with fluorine, chlorine and trifluoromethyl groups appeared to be more effective inhibitors of human and porcine aminopeptidases than unsubstituted compound **1a**. The most active compounds of the series (**1i**, **1l** and **1t**) may be considered as interesting building blocks for the synthesis of novel inhibitors of aminopeptidases. Computer-aided modeling of their interactions with both enzymes have shown that their aminophosphonate fragments are uniformly bound to the active sites of both aminopeptidases, whereas binding of the aromatic portions of inhibitory molecules to the spacious hydrophobic pockets S1 and S1′ is dependent of their substitution patterns and weakly dependent on the absolute molecular configuration.

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
