# Peer review of "Phosphonic Acid Analogs of Fluorophenylalanines as Inhibitors of Human and Porcine Aminopeptidases N: Validation of the Importance of the Substitution of the Aromatic Ring"

_biomolecules, 2020, doi:10.3390/biom10040579_

Round 1

Reviewer 1 Report

This is a very nicely written article and a very interesting work. Due to the well reported synthesis and the crystal structure data, I favor the publication of it in its current form under current circumstances. To make the story more interesting for the general audience, I suggest the following work to be conducted either as a new article or with an addition correction note in the future when the university laboratories are open again.

  1. pKa1 and pKa2 values of previously reported and the current fluorinated phenylalanine analogues should be calculated. Addition of electronegative halogens are supposed to generate an inductive effect which will alter the acidity of the end products. A table summarizing this information would be useful.
  2. In a similar way, metal binding affinities should also be reported to see the difference between fluorinated and non-fluorinated analogues and also to see the zinc binding affinity with respect to the other biologically significant metal ions such as copper, calcium, magnesium, potasium and sodium.
  3. For example, phenylphoshonic acid pKa values are between ca. 1.7 and 7.44 in general. Therefore, buffers working below pH 7.4 might not give sufficient information about the actual inhibitory activity. Buffers, that work at physiological pH would be a better option. Therefore, current data should be improved in the upcoming text. Above, pH 7.4 phosphonic acid moieties will be double deprotonated and it will result in totally different metal binding affinities (a different story). This should be addressed with an addition correction note later or with a new article.

Reviewer 2 Report

This manuscript describes the design, synthesis, and biological evaluation of a series of alanine aminopeptidase inhibitors with potential therapeutic applications, such as, in cancer treatment. 

The compounds herein reported are novel and belong to the chemical class of phosphonic  analogues of phenylalanine. 

The chemical characterization of the new compounds is complete and includes 1D/2D homo/heteronuclear NMR analysis, together with HRMS measurements. Purity of the new compounds was properly assessed by HPKC methods.

Some of these inhibitors displayed submicromolar potencies against the human isoform of the target enzyme (hAPN).

Overall, this manuscript should be published in Biomolecules after addressing the following issues:

1) The effect of chirality on the biological activity of these compounds was only hypothesized by molecular modeling studies. However, the authors should assess if there is any significant experimental difference in the activity of two enantiomers, at least for one of the most active inhibitors.

2) The description of the chemical synthesis in the experimental section should be divided for each synthetic step. They are all general procedures, but each of them should be associated to the corresponding intermediate (compounds 3/4, 5, 6/7, 1) separately.
